# Occluded Video Instance Segmentation: Dataset and ICCV 2021 Challenge

**Jiyang Qi**[1,2*] **Yan Gao**[2*] **Yao Hu**[2] **Xinggang Wang**[1] **Xiaoyu Liu**[2] **Xiang Bai**[1]
**Serge Belongie**[3] **Alan Yuille**[4] **Philip H.S. Torr**[5] **Song Bai**[2,5†]

[1]Huazhong University of Science and Technology  [2]Alibaba Group  [3]Cornell University
[4]Johns Hopkins University  [5]University of Oxford

## Abstract

Although deep learning methods have achieved advanced video object recognition performance in recent years, perceiving heavily occluded objects in a video is still a very challenging task. To promote the development of occlusion understanding, we collect a large-scale dataset called OVIS for video instance segmentation in the occluded scenario. OVIS consists of 296k high-quality instance masks and 901 occluded scenes. While our human vision systems can perceive those occluded objects by contextual reasoning and association, our experiments suggest that current video understanding systems cannot. On the OVIS dataset, all baseline methods encounter a significant performance degradation of about 80% in the heavily occluded object group, which demonstrates that there is still a long way to go in understanding obscured objects and videos in a complex real-world scenario. To facilitate the research on new paradigms for video understanding systems, we launched a challenge based on the OVIS dataset. The submitted top-performing algorithms have achieved much higher performance than our baselines. In this paper, we will introduce the OVIS dataset and further dissect it by analyzing the results of baselines and submitted methods. The OVIS dataset and challenge information can be found at `http://songbai.site/ovis`.

## 1 Introduction

In real-world scenes, visual objects are more or less occluded by other stuff. Although human vision systems can locate and recognize severely occluded objects with temporal context reasoning and prior knowledge [17, 28], it's still very challenging for current video understanding systems to perceive objects in the heavily occluded video scenes.

To facilitate future research on occlusion reasoning, we collect a large-scale dataset named **OVIS** (**O**ccluded **V**ideo **I**nstance **S**egmentation), which is specially designed for video instance segmentation in occluded scenes. The video instance segmentation task [39] requires simultaneously detecting, segmenting, and tracking all instances in a video. We believe that the videos and annotations in occlusion scenes provided by OVIS can better reveal the complexity of real-world scenes and help the development of researches in complex video understanding.

As the second video instance segmentation dataset after YouTube-VIS [39] with 232k masks, OVIS contains 296k high-quality masks and 901 complex video scenes with various occlusions. Example video clips are given in Fig 1. The most distinctive property of the OVIS dataset is that it primarily collects the videos wherein objects are under various occlusions caused by different factors. We also

---

[*]indicates equal contributions.

[†]Corresponding author. E-mail: songbai.site@gmail.com

35th Conference on Neural Information Processing Systems (NeurIPS 2021) Track on Datasets and Benchmarks.

Figure 1: Example video clips from OVIS. Watch the animations by **clicking** them (Not all PDF readers support playing animations. Best viewed with Acrobat/Foxit Reader). The annotation quality of OVIS is very high. The hairs and whiskers of animals are all exhaustively annotated.

annotate the occlusion level of each object in each frame, and present occlusion associated metrics which can measure the performance under different occlusion degrees. Therefore, OVIS will be a valuable testbed to promote future research on occlusion understanding and evaluate the performance of video instance segmentation methods on coping with occlusions.

We also evaluate nine open-source state-of-the-art methods (including FEELVOS [32], IoUTracker+ [39], MaskTrack R-CNN [39], SipMask [8], STEm-Seg [2], STMask [22], TraDeS [37], CrossVIS [40], and QueryInst [12]) on OVIS to analyze the OVIS dataset and serve as baselines for the OVIS challenge and future research. Experiment results show that our current video understanding systems fall far behind human beings in occlusion perception. The highest AP achieved by the newly proposed baselines is only 16.3, and its AP on the heavily occluded object group is only 5.6. In this sense, we still have a long way to go before deploying those techniques into practical applications, especially considering the complexity and diversity of real-world scenes.

In order to promote the research on occluded video instance segmentation, we launched the Occluded Video Instance Segmentation Challenge based on the OVIS dataset. We believe the OVIS dataset and challenge can encourage researchers to explore new video understanding systems to alleviate the occlusion issue and refresh the state-of-the-art. In Sec. 4, we review the challenge and analyze several submitted methods to dissect the OVIS dataset and serve as a reference for future research.

In summary, our main contribution are three-fold:

- We advance occlusion understanding and video instance segmentation task by releasing a new large-scale benchmark dataset named **OVIS** (short for **O**ccluded **V**ideo **I**nstance **S**egmentation). Designed with the philosophy of perceiving object occlusions in videos, OVIS could better reveal the complexity and the diversity of real-world scenes.

- We streamline the research over the OVIS dataset by conducting a thorough evaluation of nine state-of-the-art video instance segmentation algorithms, which could be a baseline reference for the OVIS challenge and future research on OVIS.

- We conduct a detailed analysis of the submitted algorithms in OVIS challenge, including their performance under different degrees of occlusion, as well as comparisons with related baselines.

## 2 Related Work

Our work focus on **Video Instance Segmentation** task in occluded scenes. The most relevant work to ours is [39], which formally defines the concept of video instance segmentation and also launches the first dataset for this task called YouTube-VIS, based on the video object segmentation dataset YouTube-VOS [38]. The YouTube-VIS dataset initially contains 2,883 videos with 4,883 instances and 131k annotated masks. In their latest 2021 version, YouTube-VIS is further extended to 3859 videos with 8,171 instances and 232k masks. As the second large-scale dataset for video instance

segmentation, OVIS focuses on the occluded scenes and mainly contains heavily occluded, more crowded, and longer instances. Therefore OVIS could better reveal the complexity of real-world scenes and help the development of research in complex video understanding.

A number of algorithms have been proposed for video instance segmentation task after the release of YouTube-VIS. MaskTrack R-CNN [39] is the first baseline method for this task. Based on the classic image instance segmentation method Mask R-CNN [16], MaskTrack R-CNN additionally predicts an embedding vector for each instance to perform tracking. MaskProp [4] is also a video extension of Mask R-CNN which first propagates the predicted masks of each frame to adjacent frames and then matches the clip-level masks for tracking. Similar to MaskTrack R-CNN, SipMask [8] also directly adds a fully-convolutional tracking branch to extend single-stage image instance segmentation to the video level. STMask [22] improves feature representation by spatial feature calibration and inferring instance masks from adjacent frames. Different from those top-down methods, STEm-Seg [2] proposes a bottom-up architecture, which performs video instance segmentation by clustering the pixels of the same instance. Built upon Transformers, VisTR [35] views the VIS task as a parallel sequence prediction problem and segments instances at the sequence level. QueryInst [12] follows a multi-stage paradigm and leverages the intrinsic one-to-one correspondence in queries across different stages. Based on the image-level instance segmentation method CondInst [31], CrossVIS [40] proposes the crossover learning scheme that uses the instance feature in the current frame to segment the same instance in other frames. Instead of following the widely-used tracking-by-detection paradigm, TraDeS [37] integrates tracking cues to assist detection.

There are also some works focusing on **occlusion understanding**. [34, 41] propose new loss functions to enforce predicted box to locate compactly to the corresponding ground-truth objects while far from other objects. [25] introduces adaptive-NMS which adaptively increases the NMS threshold in crowd scenes. [36] aggregates the temporal context to enhance the feature representations. [9] predicts multiple instances in one proposal. [18] additionally predicts the segmentation masks of occluders. [21] integrates compositional models and deep convolutional neural networks into a unified model which is more robust to partial occlusions.

Furthermore, our work is also relevant to the datasets of several other tasks, including:

**Video Object Segmentation.** Being a popular task in the video understanding area, video object segmentation (VOS) can be divided into semi-supervised VOS and unsupervised VOS according to the required supervision level at test time. Specifically, semi-supervised VOS requires tracking and segmenting a given object with its mask in the first frame, while unsupervised VOS requires segmenting the salient objects in a video without any manual annotations. As the first dataset specially designed for video object segmentation, DAVIS [29] initially contains 50 videos, and only one instance per video is annotated. While in their following works and challenges [6, 7, 30], DAVIS was extended to 150 videos with 376 densely annotated objects, and multi-object setting and interactive setting are also added. Then [38] further propose a larger dataset called YouTube-VOS. Building upon the large-scale video classification dataset YouTube-8M [1], YouTube-VOS contains 4,453 video clips and 7,755 objects. Compared with the VIS task, VOS does not distinguish semantic categories, and only one or several but not all objects are required to be segmented and tracked.

**Multi-Object Tracking and Segmentation.** Multi-Object Tracking and Segmentation task [33] (MOTS) extends the bounding box level annotation of multi-object tracking task by segmentation masks. Paul *et al.* [33] further release the KITTI MOTS and MOTSChallenge dataset by annotating the segmentation masks of KITTI tracking dataset [13] and MOTChallenge dataset [27] respectively. Different from the video instance segmentation task, MOTS focuses on the pedestrians and cars in the street scenes.

**Video Panoptic Segmentation.** Dahun *et al.* [19] extends the image-level panoptic segmentation [20] to the video domain, which requires generating consistent panoptic segmentation, and in the meantime, associating instances across frames. The reformatted VIPER dataset and the proposed Cityscapes-VPS dataset respectively contain 124 and 500 videos.

**Video Semantic Segmentation.** Video semantic segmentation is also directly extended from the image-level semantic segmentation task. Cityscapes [10] dataset contains 5000 video clips. Each clip in it consists of 30 frames and only the 20th frame of each clip is annotated. CamVid [5] dataset contains 4 videos and annotates one frame every 30 frames obtaining 800 annotated frames finally.

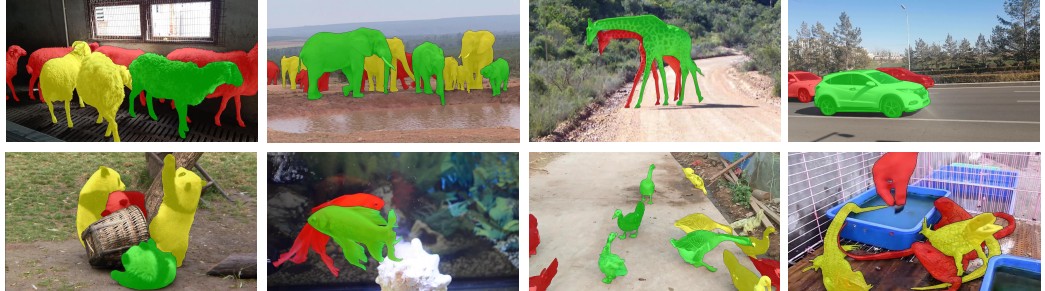

Figure 2: Annotation examples of different occlusion levels in OVIS. Green, yellow, and red respectively represent no occlusion, slight occlusion, severe occlusion.

# 3 OVIS Dataset

In this section, we will describe the collection and annotation process of OVIS, and analyze the dataset statistics.

## 3.1 Video Collection

We begin with selecting a set of semantic categories following these criteria: 1) most selected categories should be animals or vehicles, with which occlusions and movement usually occur, 2) these categories should be commonly seen in our life to reduce the difficulty of collection, 3) these categories should have a high overlap with the popular large-scale image instance segmentation datasets [14, 24] so that models trained on those image datasets will be easier to be transferred. With these criteria in mind, 25 categories are chosen, including *Person*, *Bird*, *Cat*, *Dog*, *Horse*, *Sheep*, *Cow*, *Elephant*, *Bear*, *Zebra*, *Giraffe*, *Poultry*, *Giant panda*, *Lizard*, *Parrot*, *Monkey*, *Rabbit*, *Tiger*, *Fish*, *Turtle*, *Bicycle*, *Motorcycle*, *Airplane*, *Boat*, and *Vehicle*.

As the dataset is to facilitate future research on occlusion understanding in complex scenes, we exclude 1) the videos with only one object, 2) the videos with a clean background, 3) the videos in which the complete contour of objects is visible all the time, 4) the videos in which most objects are standing still without moving. There are also some other objective rules including: 1) the length of each video is generally 5 to 60 seconds, and 2) the resolution of videos is generally $1920 \times 1080$. Besides, to preserve enough motion and occlusion scenarios, we prefer longer videos. After applying the objective rules above, only 901 video clips from the 8,644 video candidates are qualified and accepted by us.

## 3.2 Annotation

Given a qualified video, we exhaustively annotate the categories, masks, and instance identities of all the objects belonging to the pre-defined category set. Besides the common criteria (*e.g.*, no ID switch, mask fitness no more than one pixel), the annotation team is also trained with several rules particularly about occlusions: 1) if an existing object disappears because of full occlusions and then re-appears, the object identity should keep the same; 2) a new instance appeared in an in-between frame should be assigned a new object identity; and 3) the case of "object re-appears" and "new instances" should be distinguishable after watching the adjacent frames therein. We annotate all the videos every 5 frames, with the resulting annotation granularity ranges from 3 to 10 fps.

To further analyze the performance of methods under different occlusion degrees, we also annotate the occlusion degrees of each object in every frame. We divide the occlusion degrees into three levels: objects fully visible are annotated as no occlusion, objects with more than 50% visible are annotated as slight occlusion, and objects with less than 50% visible are annotated as severe occlusion. Some annotation examples are given in Fig 2. Given the annotated occlusion degree of each object in each frame, we can further get the video-level occlusion degree scores of each object. We first map the predefined three occlusion levels into numeric scores. No occlusion, slight occlusion, and severe occlusion are respectively mapped into 0, 0.25, and 0.75. Then, for an instance that appears in

| Dataset | YouTube-VIS 2019 | YouTube-VIS 2021 | OVIS (ours) |
|---|---|---|---|
| Masks | 131k | 232k | 296k |
| Instances | 4,883 | 8,171 | 5,223 |
| Videos | 2,883 | 3,859 | 901 |
| Categories | 40 | 40 | 25 |
| Video duration$^\star$ | 4.61s | 5.03s | 12.77s |
| Instance duration | 4.47s | 4.73s | 10.05s |
| mBOR$^\star$ | 0.07 | 0.06 | 0.22 |
| Objects / frame$^\star$ | 1.57 | 1.95 | 4.72 |
| Instances / video$^\star$ | 1.69 | 2.10 | 5.80 |

Table 1: Comparison of OVIS with YouTube-VIS regarding several basic or high-level statistics. See Eq. (1) for the definition of mBOR. $\star$ means the value of YouTube-VIS is estimated from the training set.

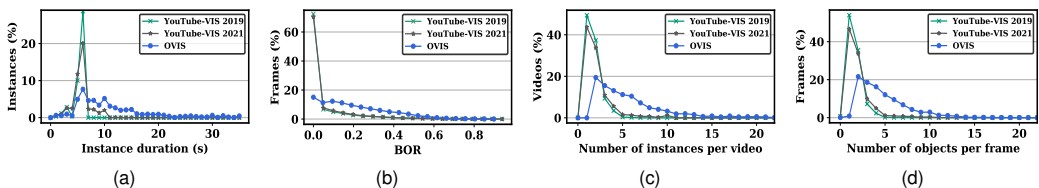

(a)          (b)          (c)          (d)

Figure 3: Comparison of OVIS with YouTube-VIS, including the distribution of instance duration (a), BOR (b), the number of instances per video (c), and the number of objects per frame (d).

multiple frames, we utilize the averaged occlusion score of the top 50% frames with highest scores to characterize the (video-level but not frame-level) occlusion degree of this instance.

Each video is firstly annotated by one annotator to get the initial annotations, and then the initial annotations are passed to another annotator to examine and correct if necessary. The final annotations are further checked by our research team.

It should be noted that while OVIS is designed for video instance segmentation, it is also suitable for semi-supervised or unsupervised video object segmentation tasks, and object tracking task is also supported as the bounding box annotations are provided. We will explore these relevant experimental settings as part of our future work.

### 3.3 Dataset Statistics

We analyze the data statistics of our OVIS dataset with YouTube-VIS 2019 and YouTube-VIS 2021 as a reference in Table 1. Note that as the annotations of only the training set are publicly available in YouTube-VIS, some statistics marked with $\star$ of YouTube-VIS are calculated only using the training set. Nonetheless, as the training set occupies 78% of the whole dataset, these statistics could still roughly reflect the properties of YouTube-VIS.

As is shown, OVIS contains 296k instance masks, which is larger than YouTube-VIS 2019 and 2021 that have 131k and 232k masks respectively. The number of instances in OVIS is larger than that in YouTube 2019 but less than that in YouTube 2021. Nonetheless, OVIS has fewer videos than YouTube-VIS as our design philosophy favors long videos and instances so as to preserve enough motion and occlusion scenarios. The number of instances per category in OVIS is also presented in Fig 4(a).

As for the length of videos and instances, the average video duration and the average instance duration of OVIS are 12.77s and 10.05s respectively, which is much longer than YouTube-VIS. The distributions of instance duration are further compared in Fig 3(a). In addition, the long range of video length can increase the diversity of OVIS, and the long videos and instances request models to have the long-term tracking ability.

In terms of occlusion degree, the proportions of objects under no occlusion, slight occlusion, and severe occlusion in OVIS are 18.2%, 55.5%, and 26.3% respectively. Besides, 80.2% of instances are

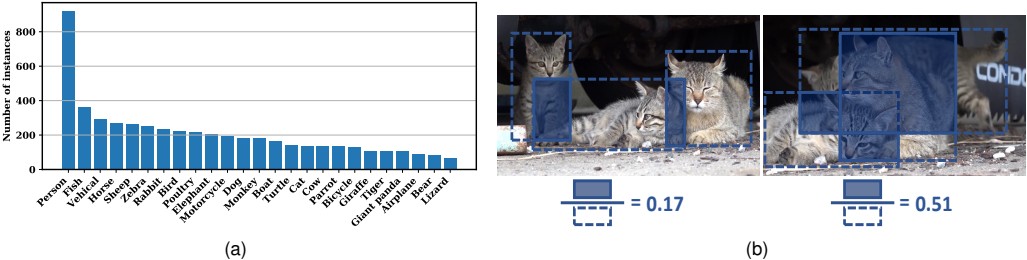

Figure 4: The number of instances per category in the OVIS dataset.

severely occluded in at least one frame, and only 2% of the instances are fully visible throughout the whole video. It supports the focus of our work, that is, to collect a dataset full of occlusion scenarios and promote the development of occlusion perception.

To further compare the degree of occlusion with other datasets and analyze the overall occlusion degree of OVIS, we define a metric named Bounding-box Occlusion Rate (BOR) to approximate the occlusion degree with only bounding box annotation. Given a video frame with $N$ objects, we denote the bounding boxes of them as $\{\mathbf{B}_1, \mathbf{B}_2, \ldots, \mathbf{B}_N\}$ and compute the BOR for this frame as

$$\text{BOR} = \frac{|\bigcup_{1 \leq i < j \leq N} \mathbf{B}_i \bigcap \mathbf{B}_j|}{|\bigcup_{1 \leq i \leq N} \mathbf{B}_i|}, \tag{1}$$

where the numerator is the area sum of the intersection between any two or more bounding boxes (in other words, we exclude those positions which only appear in an individual bounding box). The denominator is the area of the union of all the bounding boxes. An illustration is given in Fig. 4(b), showing that the heavier the occlusion is, the larger the BOR value is. While it should be mentioned here that although BOR could serve as an effective indicator for occlusion degrees using only bounding box annotations, it can only reflect the occlusion degree in a partial or rough way.

Given the BOR values of all frames, we calculate the average value of them (mBOR) to characterize the dataset in terms of the occlusion degree. Frames that do not contain any objects are ignored. As presented in Table 1, the mBOR of OVIS is 0.22, much higher than that of YouTube-VIS 2019 and YouTube-VIS 2021 (0.07 and 0.06, respectively). Fig. 3(b) further presents the distribution of BOR values, which shows that the BORs of about 70% frames in YouTube-VIS are zero. While in comparison, about half frames in OVIS locate in the region where BOR $\geq 0.2$. This suggests that there are more serious occlusions in OVIS than YouTube-VIS.

In addition to long videos and severe occlusions, OVIS also features crowded scenes, in which heavy occlusions usually occur. As shown in Table 1, there are 5.80 instances per video and 4.72 objects per frame in OVIS, while those two values are 1.69 and 1.57 in YouTube-VIS 2019, 2.10 and 1.95 in YouTube-VIS 2021. The distributions of them are further given in Fig. 3(c) and Fig. 3(d).

### 3.4 Evaluation Metrics

Following previous methods [24, 39], we utilize the average precision (AP) at different intersection-over-union (IoU) thresholds and average recall (AR) to evaluate the per-category performance of methods on OVIS. Then the mean value of APs (mAP) will be calculated as the main evaluation metric.

In addition, the annotations of occlusion levels in OVIS enable us to further evaluate the performance under different occlusion degrees. Specifically, we divide all instances into three groups called slightly occluded, moderately occluded, and heavily occluded, in which the occlusion scores (described in Sec. 3.2) of instances locate in the range of $[0, 0.25]$, $[0.25, 0.5]$, $[0.5, 0.75]$ respectively. The proportions of the three groups are 23%, 44%, and 49% respectively. Thereby, we can calculate the AP under each occlusion level (denoted by $\text{AP}_{SO}$, $\text{AP}_{MO}$, and $\text{AP}_{HO}$ respectively) by ignoring the instances of other levels.

| Methods | OVIS validation set | | | | | | | | OVIS test set | | | | | | | |
|---|---|---|---|---|---|---|---|---|---|---|---|---|---|---|---|---|
| | AP | $AP_{50}$ | $AP_{75}$ | $AR_1$ | $AR_{10}$ | $AP_{SO}$ | $AP_{MO}$ | $AP_{HO}$ | AP | $AP_{50}$ | $AP_{75}$ | $AR_1$ | $AR_{10}$ | $AP_{SO}$ | $AP_{MO}$ | $AP_{HO}$ |
| FEELVOS [32] | 9.6 | 22.0 | 7.3 | 7.4 | 14.8 | 17.3 | 11.5 | 1.7 | 10.8 | 23.4 | 8.7 | 9.0 | 16.2 | 18.9 | 12.2 | 2.0 |
| IoUTracker+ [39] | 7.0 | 16.9 | 5.3 | 5.7 | 14.3 | 11.5 | 7.9 | 1.8 | 8.0 | 18.4 | 7.5 | 5.9 | 15.7 | 12.8 | 9.1 | 2.1 |
| MaskTrack R-CNN [39] | 10.8 | 25.3 | 8.5 | 7.9 | 14.9 | 23.0 | 12.8 | 2.7 | 11.8 | 25.4 | 10.4 | 7.9 | 16.0 | 22.7 | 15.0 | 3.5 |
| SipMask [8] | 10.2 | 24.7 | 7.8 | 7.9 | 15.8 | 19.9 | 10.5 | 2.2 | 11.7 | 23.7 | 10.5 | 8.1 | 16.6 | 21.9 | 13.9 | 3.2 |
| STEm-Seg [2] | 13.8 | 32.1 | 11.9 | 9.1 | 20.0 | 22.2 | 16.1 | 3.9 | 14.4 | 30.0 | 13.0 | 10.1 | 20.6 | 22.5 | 16.8 | 4.2 |
| TraDeS [37] | 11.4 | 26.5 | 9.4 | 7.0 | 13.8 | 23.0 | 12.8 | 3.0 | 12.0 | 26.4 | 10.8 | 7.8 | 14.6 | 21.6 | 14.1 | 3.6 |
| QueryInst-VIS [12] | 14.7 | **34.7** | 11.6 | 9.0 | 21.2 | 27.3 | 17.2 | 4.1 | 16.0 | **33.7** | 14.7 | 9.6 | 21.7 | 26.3 | 17.7 | 4.5 |
| STMask [22] | **15.4** | 33.8 | **12.5** | 8.9 | **21.3** | 24.0 | **18.7** | **5.1** | 15.6 | 32.5 | 13.8 | 9.1 | **21.8** | 25.4 | 17.1 | **6.3** |
| CrossVIS [40]* | 14.9 | 32.7 | 12.1 | **10.3** | 19.8 | **28.4** | 16.9 | 4.1 | **16.3** | 31.5 | **15.4** | **10.6** | 21.1 | **27.3** | **18.5** | 5.6 |
| Ach [23] | 28.9 | 56.3 | 26.8 | 13.5 | 34.0 | 45.3 | 31.9 | 12.9 | 32.2 | 56.3 | 33.3 | 15.5 | 36.7 | 44.6 | 35.9 | 15.4 |
| Ali2500 [3] | 21.3 | 43.9 | 18.8 | 13.3 | 28.5 | 35.6 | 25.1 | 5.7 | 21.6 | 39.8 | 20.2 | 12.6 | 27.4 | 34.1 | 26.0 | 6.0 |
| LI-Minghan [22] | 19.7 | 39.8 | 17.0 | 10.8 | 26.7 | 32.3 | 24.0 | 5.7 | - | - | - | - | - | - | - | - |

Table 2: Overall results of nine baseline methods and three submitted methods on the OVIS dataset. $AP_{SO}$, $AP_{MO}$, and $AP_{HO}$ respectively denote the AP of objects slightly occluded, moderately occluded, and heavily occluded. * means the baseline model is additionally pre-trained with the YouTube-VIS dataset [39]. The AP scores of all baselines we run are within a margin of deviation of $\pm 0.7$.

# 4 OVIS Challenge

To further encourage the exploration of new paradigms for video understanding, we launched the Occluded Video Instance Segmentation Challenge. In this section, we will report and analyze the results of a number of baseline methods and the submitted approaches, as a reference for future research on OVIS and occlusion understanding.

## 4.1 Competition Overview

We divide the newly collected OVIS dataset into 607 training videos, 140 validation videos, and 154 test videos, as our official split. For each category, there are at least 4 videos in each of validation set and test set. And the split proportions of different categories are guaranteed to be approximately the same.

There were a total of 163 participates registered for the competition and 7 teams submitted the final results on the test split. We will choose three representative methods to analyze in Sec. 4.3.

## 4.2 Baselines

To provide baseline references to the OVIS challenge and future research, we evaluated 9 existing state-of-the-art methods on OVIS, including mask propagation methods (*e.g.*, FEELVOS [32]), track-by-detect methods (*e.g.*, IoUTracker+ [39]), and recently proposed end-to-end methods (*e.g.*, Mask-Track R-CNN [39], SipMask [8], STEm-Seg [2], STMask [22], TraDeS [37], CrossVIS [40], and QueryInst [12]). The segmentation mask of the first frame given to FEELVOS are predicted by Mask R-CNN but not the ground truth. All our baselines adopt ResNet-50-FPN [15] as the backbone. CrossVIS is pre-trained on both MS-COCO [24] and YouTube-VIS [39] while all other methods are initialized with parameters only pre-trained on COCO [24]. Input frames are downsampled to $640 \times 360$ in both training and inference following previous works [39]. We conduct all our experiments with four V100 GPUs. Except for STMask and CrossVIS whose results are evaluated with the checkpoints provided by their authors, the reported results of all the other baselines are the averages of three runs.

Despite most of baselines can obtain more than 30 AP on the YouTube-VIS dataset (the results of baselines on YouTube-VIS 2019 validation set are presented in Appendix B), all baseline methods suffer from a performance drop of at least 50% on OVIS compared with that on YouTube-VIS, as shown in Table 2. Especially when evaluating on the heavily occluded instance group, all methods encounter a significant performance degradation. For example, while achieving an AP of 32.5 on YouTube-VIS, SipMask [8] only obtains an AP of 11.7 on the OVIS test split and a much lower AP of 3.2 on the heavily occluded group. It firmly suggests that severe occlusion will greatly improve the difficulty of video instance segmentation, and current video understanding systems are not satisfying. Especially considering the complexity and diversity of scenes in the real visual world, further attention should be paid to video instance segmentation in the real world where occlusions extensively happen.

| Methods | Backbone | OVIS validation set | | | | | | | |
|---------|----------|-----|------|------|------|------|------|------|------|
| | | AP | $AP_{50}$ | $AP_{75}$ | $AR_1$ | $AR_{10}$ | $AP_{SO}$ | $AP_{MO}$ | $AP_{HO}$ |
| MaskTrack R-CNN [39] | ResNet-50 | 10.8 | 25.3 | 8.5 | 7.9 | 14.9 | 23.0 | 12.8 | 2.7 |
| MaskTrack R-CNN [39]+LSS | ResNet-50 | 13.5 | 29.9 | 11.3 | 8.5 | 18.7 | 25.4 | 16.7 | 3.3 |
| MaskTrack R-CNN [39]+LSS | Swin-S | 21.1 | 42.1 | 20.0 | 12.2 | 26.8 | 38.2 | 23.8 | 6.6 |
| Ach w/o SWA [23] | Swin-L | 28.0 | 56.5 | 25.8 | 13.6 | 33.1 | 43.9 | 32.1 | 13.0 |
| Ach w/ SWA [23] | Swin-L | 28.9 | 56.3 | 26.8 | 13.5 | 34.0 | 45.3 | 31.9 | 12.9 |
| STEm-Seg [2] | ResNet-50 | 13.8 | 32.1 | 11.9 | 9.1 | 20.0 | 22.2 | 16.1 | 3.9 |
| STEm-Seg [2]+image data | ResNet-50 | 16.2 | 36.2 | 13.2 | 10.8 | 22.7 | 27.4 | 18.5 | 4.4 |
| Ali2500 [3] | ResNeXt-101 | 21.3 | 43.9 | 18.8 | 13.3 | 28.5 | 35.6 | 25.1 | 5.7 |

Table 3: Results of some ablation experiments on the approach of team Ach and Ali2500. The "Ach w/o SWA" is approximately equivalent to "MaskTrack R-CNN+LSS" with Swin-L backbone, while some hyperparameters (*e.g.*, input resolution, training schedule) between them are different. "+image data" means additionally generating image pairs from COCO [24] for training, as team Ali2500 and [2] did. The "Ali2500" is approximately equivalent to "STEm-Seg+image data" with ResNeXt-101 backbone.

Although all the baselines perform not so satisfying on OVIS, there are still some baselines (*e.g.*, STMask [22], STEm-Seg [2]) that achieve relatively higher performance on the heavily occluded object group, which demonstrate that the architecture or pipelines of these methods might more suitable to heavy occlusion perception.

### 4.3 Approaches

In this sub-section, we will briefly review the methods of three representative submissions and analyze their results with the reference of baselines. The final results of them are presented in Table 2.

**Team Ach [23].** Team Ach develop their approach based on MaskTrack R-CNN [39]. Considering the videos in OVIS are so long that the randomly sampled reference frame in MaskTrack R-CNN might be quite different from the query frame, they propose a new sampling strategy that only samples the reference frame from the neighboring $n = 5$ frames. To further improve the performance, they adopt Swin-L [26] as the backbone and apply stochastic weights averaging (SWA) training strategy. Larger input resolution is also applied. Finally, they obtain an AP of 28.9 on the validation set and an AP of 32.2 on the test set, significantly outperforming all the baselines.

To further figure out the AP improvement of each modification, we also conduct several additional ablation experiments. As shown in Table 3, the long videos in OVIS make the final performance more sensitive to the sampling strategy of reference frame during training. By replacing the naïve random sampling with the proposed limited sampling strategy, MaskTrack R-CNN can achieve a remarkable AP improvement of 2.7. Besides, the Swin transformer backbone can greatly boost the AP (from 13.5 to 21.1 by only replacing ResNet-50 with Swin-S) and even the $AP_{HO}$ is also significantly improved, which suggests that finer features extracted by powerful backbone can largely alleviate the occlusion issue. The SWA training strategy can also bring an AP improvement of 0.9.

Furthermore, we also visualize the predicted results of team Ach [23] on several video clips in Fig. 5, to present the perception problems caused by occlusion. In (a), two persons and two bicycles heavily overlap with each other. The model successfully tracks the bicycles but fails to track the person. In (b), when two bears are intersecting, severe occlusion leads to failure of detection and tracking. In (c), the fish colored green is well tracked, but the fish colored blue is failed to be re-tracked after being fully occluded by the purple fish. In (d), the model fails to recognize the heavily occluded yellow airplane in the 4th frame. Besides, when the airplanes are very close to each other, the model is confused and doesn't know which one to segment. In (e), the ID switch error is also encountered in the 3rd frame when intersecting. And in the 4th frame, the bounding box of a monkey is suppressed due to occlusion.

**Team Ali2500 [3].** Team Ali2500 build their approach based on the bottom-up method STEm-Seg [2]. Benefiting from 3D convolutional layers and the bottom-up architecture, STEm-Seg baseline surpasses many methods on OVIS and obtains a relatively high $AP_{HO}$ of 4.2 on the test split. We guess this is because that 3D convolution can model the temporal context more effectively, and the bottom-up architecture avoids the detection process which is difficult in occluded scenes.

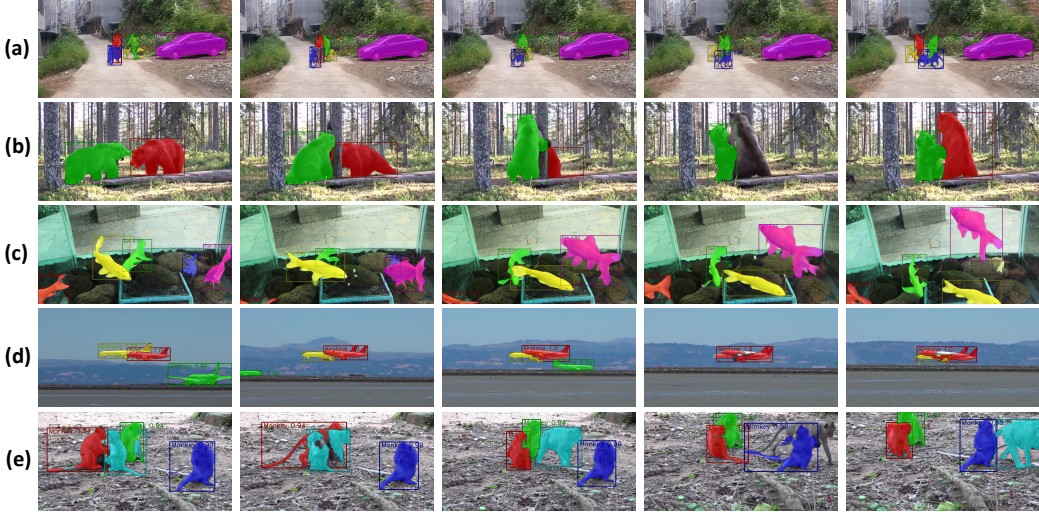

Figure 5: Evaluation examples of team Ach on OVIS. Each row presents the results of five frames from one video.

Team Ali2500 further extends STEm-Seg with a stronger baseline (ResNeXt-101) and leverages the image-level instance segmentation dataset COCO [24] by synthesizing image pairs from the single images for training. The augmented image sequences can effectively enlarge the number of training scenes and increase the robustness of methods. Finally, they obtain an AP of 21.3 on the validation set and an AP of 21.6 on the test set.

Following [2], we also conduct an ablation experiment to figure out the performance improvement of additionally training with synthesized image pairs on OVIS. The results are presented in Table 3. By additionally synthesizing training image pairs from COCO [24] and Pascal VOC [11] datasets, STEm-seg achieves an AP of 16.2, a remarkable AP improvement of 2.4 over its original baseline.

**Team LI-Minghan [22].** Thanks to the temporal fusion module, STMask [22] can complement the missing object cues caused by occlusion with the reference of adjacent frames, which enables it to outperform other baselines on perceiving severely occluded objects (as shown in Table 2). By adopting the stronger ResNet-152 backbone, team LI-Minghan further improves the AP of STMask from 15.4 to 19.7 and achieves an $AP_{HO}$ of 5.7 on the validation split.

## 5 Limitations

Our work is a valuable benchmark for occlusion reasoning and video instance segmentation, but there are still several limitations that need to be addressed in future work.

First, as our design philosophy favors long and crowded videos to preserve enough occlusions in one video, OVIS contains fewer videos than some previous datasets (*e.g.*, YouTube-VIS [39]), which may reduce the variance in scenes and affect the generalization capability of methods trained on OVIS. And considering the data inadequacy problem caused by high costs of annotation is common among most video segmentation datasets, exploring how to leverage the large-scale image-level instance segmentation datasets and exploiting unlabeled data will be meaningful.

In addition, the poor performance of methods on the OVIS dataset has demonstrated that we are still at a nascent stage for understanding objects, instances, and videos in a real-world occluded scenario. More effort should be devoted in the future to tackling object occlusions by contextual reasoning or associating.

## 6 Conclusion

In this paper, we mainly introduce the OVIS dataset, which is specially designed for video instance segmentation in occlusion scenes. OVIS consists of 296k high-quality masks of 5,223 severely

occluded instances. To dissect the OVIS dataset and facilitate future research on occlusion perception, we also conduct a comprehensive evaluation of nine baselines and three submitted methods in the OVIS challenge, which can be a reference for future work. The unsatisfying performance on OVIS suggests that more attention should be paid to real-world scenario understanding. For future works, we plan to extend the OVIS dataset to some relevant tasks, such as semi-supervised/unsupervised video object segmentation or video panoptic segmentation [19]. We believe the OVIS dataset can be a useful testbed and inspire more research in understanding videos in complex and diverse scenes.

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
