# OpenReview forum: "Occluded Video Instance Segmentation: Dataset and ICCV 2021 Challenge"
_NeurIPS.cc/2021/Track/Datasets_and_Benchmarks/Round2 — NeurIPS 2021 Datasets and Benchmarks Track (Round 2)_

### Official Review · Reviewer_Mk48 · 2021-09-16
**A reasonable and meaningful dataset for video instance segmentation.**

**Rating:** 7
**Confidence:** 5
**Correctness:** Yes. The dataset looks sound and reas…
**Clarity:** Yes. The paper is well written and ea…

**Strengths:**

- A meaningful dataset that focuses on occluded video instances, which can help to better evaluate the performance of VIS/VOS models.
- A comprehensive analysis of the proposed dataset is presented.
- Several recent SOTA models are tested and compared on the proposed dataset, which is helpful for future research in this field.



**Weaknesses:**

- The size of the proposed dataset is small comparing to existing datasets like youtube-vis.
- VIS task and VOS task are somehow mixed in Tab.2. VOS models like FEELVOS require annotation in the first frame, while VIS models like MaskTracker don't need that. Hence it should be better to evaluate performance for the VOS task and the VIS task individually.

**Additional Feedback:**

No

**Documentation:**

Yes.

**Ethics:**

The authors may need to protect some private information in the dataset, e.g human faces, and vehicle license plates.

**Relation To Prior Work:**

Yes.

**Summary And Contributions:**

This work presents a video instance segmentation dataset that focuses on scenes with occlusions. The dataset has 901 videos containing 5,223 instances of 25 categories. Several recent state-of-the-art models for VIS and VOS are evaluated on this dataset, and the results demonstrate that the proposed dataset is challenging and difficult.

---

> ### Author Response · Authors · 2021-09-30
> **Thanks and Response to Reviewer Mk48**
>
> Thank you for your acknowledgment and helpful advice, and we have updated our manuscript correspondingly. The responses to the main concerns are as follows.
>
> ###### **Q1: The size of the proposed dataset is small comparing to existing datasets like YouTube-VIS.**
>
> As stated in Section 5, this is a weak point of OVIS as our design philosophy favors "long and crowded" videos to preserve enough occlusions but not "more'' videos. Nevertheless,
>
> - OVIS contains more segmentation masks, more instances per category, and more complex scenes than YouTube-VIS.
> - Most categories we selected can be found in the popular large-scale image instance segmentation datasets [1, 2], which means we can better leverage those image-level data to help models learn feature representations and improve generalization, as [4] did. In our experiments, by additionally synthesizing training frame pairs from COCO and Pascal VOC datasets, STEm-seg achieves an AP of 16.2 on OVIS, a remarkable AP improvement of 2.4 over its original baseline. The detailed results have been updated in Table 3. We will further add a new table with more baseline results trained with synthesized training clips in our final version, to encourage the exploration of leveraging image datasets to improve generalization and stability.
>
> ###### **Q2: It should be better to evaluate performance for the VOS task and the VIS task individually.**
>
> Sorry for not clarifying the implementation details of FEELVOS. Following [3], in our experiments, the segmentation masks of the first frame given to FEELVOS are predicted by Mask R-CNN but not the ground truth. By doing so, we adapt FEELVOS to the VIS task to be a baseline for mask propagation methods. We have clarified the implementation details of FEELVOS in section 4.2.
>
> ###### **Q3: The authors may need to protect some private information in the dataset, e.g. human faces, and vehicle license plates.**
>
> Thank you for pointing out this concern. We have removed the videos with degrading or embarrassing contents and clipped or blurred the parts containing identity information, e.g., names, human faces, and vehicle license plates, to protect privacy.
>
> [1] Everingham, Mark, et al. "The pascal visual object classes (voc) challenge." IJCV 88.2 (2010): 303-338.
>
> [2] Lin, Tsung-Yi, et al. "Microsoft coco: Common objects in context." In ECCV 2014.
>
> [3] Yang, Linjie, et al. "Video instance segmentation." In ICCV 2019.
>
> [4] Athar, Ali, et al. "Stem-seg: Spatio-temporal embeddings for instance segmentation in videos." In ECCV 2020.

---

### Official Review · Reviewer_D1xQ · 2021-09-19
**Very useful and well-thought out benchmark with some minor issues in analysis and benchmarking**

**Rating:** 7
**Confidence:** 3

**Strengths:**

- The dataset is very well thought out with the rationale behind different design choices clearly outlined.
- The splits and metrics chosen to analyze the models make a lot of sense and allow the reader to gain some insight into where the model might be failing or succeeding.
- The visualizations on the websites look very nice with good reidentification of the same object across the video.
- I like noting not only how the dataset is currently used, but also other ways in which it could be used with little modification.


**Weaknesses:**

1. While Table 2 lists OVIS numbers, there is very little discussion of how those methods do on prior datasets with the exception of a brief comment in L239. I think including the results on those prior methods would be very useful here.

2. One of the main claims in the paper is that prior datasets do not adequately capture the complexity of the real world. While the authors show the performance of models on their dataset, this confounds two issues: dataset set complexity and dataset size. A lot of recent work on image-based models show that more data with weaker annotation can often lead to comparable performance to less data with stronger supervision/annotation. I think cross-dataset generalization experiments could really shed a lot of light on what is going on here. Do models train on YouTube-Vis do well on OVIS and vice-versa? This would help clarify both the challenge of evaluating occluded scenes as well as the potential benefit of training on occluded scenes.

3. The authors do a great job on discussing the different models and trying to explain the reasons behind the differences in performance. However, the three representative submissions discussed in Sec 4.3 have different visual backbones, modules, and visual augmentations. I think this weakness of any analysis regarding what caused the models to improve. An ablation study similar to the one shown in Table 3 would be helpful to explain the approaches proposed by Team Ali2500 and Team L1-Minghan.

4. There is little to no discussion of such issues despite L394 indicating otherwise.

5. While 3 models are trained and averaged to get a more representative performance, there aren’t any error bars reported or variance in performance across those runs despite L409 stating otherwise.


**Additional Feedback:**

**Questions to authors**

I had two discussion questions regarding the paper that were not weaknesses or issues, but rather simply questions to the authors:

1. In L129, the authors state that they remove videos in which most objects are standing still without moving. I am curious if the dataset still contains many non moving objects. One visualization shows the existence of subject objects (bike in third row of the visualization). I am curious about this since it could result in a bias against correctly segmenting objects when they are not moving.
2. In L143, it is noted that videos range from 3 to 6 fps. Does that mean that some videos were only captured/released at 15 fps? Do you observe any impact on segmentation performance based on the annotation granularity.

---------------

**Minor issues/Presentation suggestions**

Those points are minor issues or presentation suggestions and did not impact the rating of the paper. The authors do not need to respond to them, but I would encourage them to consider them. Comments in (rough) order of appearance in paper.

- L13: typo, based not basing
- L19-22: the word our seems unnecessary here.
- Figure 1: while showing animations is great, I noticed that they do not show up at all in printed documents. I would suggest using something that is more accessible in the paper and reserving animations for online viewing.
- L147: severe no server.
- L154-156: future tense used when I think the present tense should be used.
- Table 2: AP_{HO} goes over the line.
- Table 2 and L229: I think it would be good to include the results on the VIS dataset as well instead of simply mentioning the difference in text.
- L255: typo, based not basing.
- Figure 5: I think describing the failure mode for each row in the caption or in the figure would help make this figure easier to understand. It looks good now, but I think adding a succinct summary of L270-278 here would make it much better and more self-contained.


**Clarity:**

Overall, the paper is well-written and was easy to read and comprehend. There are a few exceptions that I list below:
- It’s unclear how the average recall metrics were computed. My guess is that the subscript refers to the classes, but I would suggest clarifying it explicitly in the text.
- Table 3 was a bit unclear. Based on the text [L255-269], it seems that Ach w/o SWA is equivalent to MaskTrack R-CNN + LSS + Swin-L, is this correct? If so, I think rewriting the first column descriptions would make this figure much clearer. If not, it would be great to clarify the relation between the top 3 rows and the bottom 2. This is particularly important since the jump from row 3 to row 4 is the second largest jump for most columns and the highest in high occlusion.
- Why are the results for L1-Minghan on the test set missing?


**Correctness:**

Overall, the dataset was constructed well and the design choices make sense with their rationale clearly outlined. One issue I had was with the definition and presentation of the mBOR metric.

**Systematic issue with mBOR metric:**
If I understood the mBOR metric correctly, then I think it has a major confounder when comparing to the previous datasets.
The authors use the bounding boxes to get a pseudo-measure of object overlap and occlusion. I found this metric very interesting and creative, but it seems to hinge on the assumption that potential occluders are annotated. It’s unclear if this assumption applies equally strongly across datasets. For example, Figure 5 row b shows the bear being occluded by the tree.  This occlusion would not be counted by the mBOR metric since the tree is not annotated. While this issue applies to both YouTube VIS and OVIS, it seems from the datasets descriptions that it will systematically affect YouTube VIS more due to the number of annotated objects per frame since it’s unclear how well that correlates with number of actual objects in the frame. I understand that quantifying this is difficult, but I think the authors should discuss it in the paper to either refute my concern in case it’s correct or acknowledge that this metric will be unfair to prior works with less objects/frame annotated.


**Documentation:**

Seems fine.


**Ethics:**

There is little to no discussion of ethics or societal impact of the paper. I do not think the current paper does anything unethical, but the lack of discussion is a bit concerning.

**Relation To Prior Work:**

Work is well-positioned with respect to prior work.


**Summary And Contributions:**

The authors propose a video dataset focusing on segmenting occluded objects. This is an interesting problem as it tackles a more realistic setup where objects get occluded and disoccluded during the video instead of being clearly in view throughout the video. The authors clearly present the rationale behind their design choices and their approach towards creating this dataset. Finally, the authors compare the performance of several models on their benchmark and provide some analysis of the results and model ablations for one of the approaches.

---

> ### Author Response · Authors · 2021-09-30
> **Thanks and Response to Reviewer D1xQ (Part 2)**
>
> ###### **Q7: It’s unclear how the average recall metrics were computed.**
>
> Following previous work [1, 2], AR is defined as the maximum recall given a fixed number of segmented instances per video. The subscript of AR refers to the max number of predictions per video to be counted.
>
> ###### **Q8: Table 3 was a bit unclear.**
>
> Thank you for your advice.
>
> - "Ach w/o SWA" is approximately equivalent to "MaskTrack R-CNN + LSS + Swin-L", but not exactly. Team Ach utilized larger input resolutions, a longer training schedule, and fewer predictions per frame. As we don't have access to the code of Ach, we can't guarantee that there aren't more detailed differences. We have clarified this in the caption of Table 3.
> - The large performance gap between row 3 and row 4 is caused by the difference in backbone. Swin-L is significantly better than Swin-S. We are trying to train the "MaskTrack R-CNN + LSS + Swin-L" and we will update its results in our final paper.
>
> ###### **Q9: Why are the results for LI-Minghan on the test set missing?**
>
> Team LI-Minghan didn't submit their results for the test set.
>
> ###### **Q10: I am curious if the dataset still contains many non-moving objects.**
>
> We only exclude the videos in which "most" objects (but not "one or more objects") are standing still without moving. So there are still many non-moving or not-always-moving objects in the OVIS dataset.
>
> ###### **Q11: Does some videos were only captured/released at 15 fps? Do you observe any impact on segmentation performance based on the annotation granularity?**
>
> There are only three videos are captured at 15 fps in OVIS, while most videos are 25 or 30 fps. We didn't observe any impacts caused by the annotation granularity for most objects. While for a small number of fast-moving objects, dense annotations may greatly improve their tracking performance.
>
> [1] Lin, Tsung-Yi, et al. "Microsoft coco: Common objects in context." In ECCV 2014.
>
> [2] Yang, Linjie, et al. "Video instance segmentation." In ICCV 2019.
>
> [3] Athar, Ali, et al. "Stem-seg: Spatio-temporal embeddings for instance segmentation in videos." In ECCV 2020.

---

> ### Author Response · Authors · 2021-09-30
> **Thanks and Response to Reviewer D1xQ (Part 1)**
>
> Thank you for your detailed comments and helpful advice, and we have updated our manuscript correspondingly. The responses to the main concerns are as follows.
>
> ###### **Q1: There is very little discussion of how those methods do on prior datasets with the exception of a brief comment in L239.**
>
> Thanks for your advice. We have presented the performance of baselines on YouTube-VIS in Appendix B.
>
> ###### **Q2: The authors confound two issues: dataset set complexity and dataset size. Do models trained on YouTube-VIS do well on OVIS and vice-versa?**
>
> Considering the categories in OVIS are different from YouTube-VIS, we calculate the mAP of their 19 common categories as the metric. However, as we can't get the per-category AP scores from the YouTube-VIS evaluation server, we can only evaluate how the models trained on YouTube-VIS perform on OVIS.
>
> On these 19 categories, MaskTrack R-CNN [2] trained on OVIS and YouTube-VIS obtain an AP of 11.7 ($AP_{SO}=22.1,AP_{MO}=12.9,AP_{HO}=2.8$) and 4.7 ($AP_{SO}=7.1,AP_{MO}=5.9,AP_{HO}=0.4$) on OVIS. The large performance gap may be caused the large differences of scenes between these two datasets. Most videos in YouTube-VIS are simple with only one or two objects in it, while the scenes in OVIS are more complex and full of occlusions. And the $AP_{HO}$ of the model trained on YouTube-VIS is almost zero.
>
> To further study the impact of data amount, following [3], we evaluated the performance gain of training with synthesized clips from single images. By additionally synthesizing training frame pairs from COCO and Pascal VOC datasets, STEm-seg achieves an AP of 16.2, a remarkable AP improvement of 2.4 over its original baseline. The detailed results have been updated in Table 3. To encourage the exploration of leveraging image datasets to improve generalization and stability, we will provide more baseline results trained with synthesized training clips in our final version.
>
> ###### **Q3: An ablation study similar to the one shown in Table 3 would be helpful to explain the approaches proposed by Team Ali2500 and Team LI-Minghan.**
>
> Thank you for your advice. The analysis of team Ali2500 and LI-Minghan in Section 4.3 are all based on their corresponding baselines with ResNet-50 as the backbone. And as mentioned in Q2, we have evaluated STEm-seg's performance gain achieved by training with synthesized image pairs and presented the results in Table 3. We will also add a new table with more baseline results trained with synthesized training clips in our final version. Further ablation experiments for STMask (STMask needs about three days on two P100 GPUs to train) are also running. However, as we don't have access to the source code of these teams, these ablation experiments conducted by us may not be so rigorous.
>
> ###### **Q4: There is little to no discussion of ethics or societal impact of the paper.**
>
> We don't think there are any potential negative societal impacts or ethical concerns. We have removed the videos with degrading or embarrassing contents and clipped or blurred the parts containing identity information, e.g., names, human faces, and vehicle license plates, to protect privacy. And all video uploaders have consented to the public research usage of the videos.
>
> ###### **Q5: There aren’t any error bars reported or variance in performance across those runs.**
>
> The AP scores of all baselines we run are within a margin of deviation of $\pm 0.7$. We have added this to the caption of Table 2. Caused by data inadequacy, large variance in performance is also [a common issue](https://github.com/youtubevos/MaskTrackRCNN/issues/6#issuecomment-502503505) of video segmentation benchmarks.
>
> ###### **Q6: Systematic issue with mBOR metric. mBOR metric will be unfair to prior works with fewer objects/frames annotated.**
>
> Thank you for pointing out this concern. First of all, as stated in Section 3.3, mBOR can only reflect the occlusion degree in a partial or rough way. Objects only occluded by unannotated objects, occluded by image boundaries, or full occluded will all not be counted by mBOR. So mBOR can't be a completely fair metric. If a commonly seen occluder is annotated in one dataset but not be annotated in another dataset, mBOR may not be a good metric. While in Youtube-VIS, there is nearly no occluder that is annotated by OVIS but isn't annotated by YouTube-VIS. Therefore, we don't think mBOR is unfair to YouTube-VIS.
>
> [1] Lin, Tsung-Yi, et al. "Microsoft coco: Common objects in context." In ECCV 2014.
>
> [2] Yang, Linjie, et al. "Video instance segmentation." In ICCV 2019.
>
> [3] Athar, Ali, et al. "Stem-seg: Spatio-temporal embeddings for instance segmentation in videos." In ECCV 2020.

---

### Official Review · Reviewer_1djJ · 2021-09-21
**Important problem and major contribution**

**Rating:** 8
**Confidence:** 4
**Correctness:** The material appears to be correct.
**Clarity:** The paper is relatively well-written.

**Strengths:**

1. The authors create a large dataset of videos with annotations to support instance segmentation
2. The authors annotate occlusions in their dataset to enable evaluation of models in the presence of varying degrees of occlusion
3. The authors provide strong empirical results to support the hypothesis that occlusion is indeed an important problem to study.
4. The authors have already solicited solutions for their benchmark, and they discuss some of the best-performing models.

**Weaknesses:**

1. The authors show strong empirical evidence that the performance of contemporary instance segmentation models degrades rapidly in the presence of occlusion.  However, the authors use this to claim (e.g., in Section 5) that more work needs to be done to solve occlusion.  This assertion assumes however that better performance could be achieved with a more sophisticated model: how do we know this is true though?  And if we can achieve better performance in the presence of occlusion, then how much?  At what point can we say that the problem is approximately solved?

While I largely agree with the authors' assertion, it seems to me that it still lacks strong empirical or theoretical evidence.  This is because occlusion does degrade model performance in an irrecoverable way; occlusion reduces the information available to a recognition model (computer or human) to make a decision.  Therefore, how do we know when we achieve the best possible performance, given the occlusion present?   It would be really helpful if the authors could discus this issue, and/or provide some estimate of the "upper bound" achievable performance using a human annotator (for example).


2. It would be helpful if the authors could briefly discuss existing work on image recognition in the presence of occlusion (e.g., Compositional neural networks Kortylewski et al., 2020).  I think this would provide historical context for readers, and strengthen the case that occlusion is indeed a problem for recognition models.



**Additional Feedback:**

None.

**Documentation:**

The documentation appears complete and high quality.

**Ethics:**

I do not have any remaining ethical concerns.

**Relation To Prior Work:**

The related work is clear.

**Summary And Contributions:**

The authors create a large new video dataset for evaluating instance segmentation models in the presence of occlusion.  The authors empirically demonstrate that occlusion is an important problem by showing the substantial performance degradation of many state-of-the-art models in the presence of occlusion.  This hypothesis has been supported by other recent work in image classification (e.g., Compositional neural networks Kortylewski et al., 2020), however, this work still adds substantial value by evaluating many state-of-the-art models, and also studying this problem in the context of video instance segmentation.   Although this is an important problem, as far as I'm aware there are no benchmarks for occlusion, or even datasets that have labeled real-world occlusions that could support the evaluation of models in the presence of occlusion.  In summary, I think this is an important problem and this benchmark represents a major contribution to support research towards solving this problem.

---

> ### Author Response · Authors · 2021-09-30
> **Thanks and Response to Reviewer 1djJ**
>
> Thank you for your thoughtful comments and advice, and we have updated our manuscript correspondingly. The responses to the main concerns are as follows.
>
> ###### **Q1.1: Why we believe that better performance could be achieved with more work?**
>
> Some previous works in other tasks have shown that the performance in occluded scenes can be improved if the specially designed losses [1, 2], compositional models [4], specially designed NMS methods [3, 5], temporal context aggregation [6], occluder prediction module [7], or some other occlusion-aware modules are applied.
>
> In addition, as presented in Fig. 5, most objects are very easy for humans to perceive and track. While being the top-performing method in the challenge, Ach's model usually fails to track and segment objects in Fig. 5 when they are occluded or close to other objects, which reveals that current deep learning models perform differently with the human vision system. So, a new video understanding paradigm that can leverage more prior knowledge of human beings should be designed.
>
> ###### **Q1.2: At what point can we say that the occlusion problem is approximately solved?**
>
> Thanks for this insightful comment. As you said, heavy occlusion can irrecoverably degrade the performance of human vision systems. While, most of the time, the impacts of occlusion on human beings are minor and ignorable, compared with current video understanding systems. So, in my opinion, if one day, algorithms' error rates caused by occlusion can be equivalent to that of human beings, we can say that the occlusion problem is approximately solved. While if we ignore the impacts of occlusion on human beings, we believe the $AP_{SO}$ can be deemed as an approximate upper bound of $AP_{MO}$ and $AP_{HO}$.
>
> ###### **Q2: Brief discussion of existing work on image recognition in the presence of occlusion.**
>
> Thank you for the advice. We have updated the discussion of previous work on occlusion understanding in Section 2. The added paragraph is quoted below.
>
> > There are also some works focusing on occlusion understanding. [1, 2] propose new loss functions to enforce predicted box to locate compactly to the corresponding ground-truth objects while far from other objects. [3] introduces adaptive-NMS which adaptively increases the NMS threshold in crowd scenes. [6] aggregates the temporal context to enhance the feature representations. [5] predicts multiple instances in one proposal. [7] additionally predicts the segmentation masks of occluders. [4] integrates compositional models and deep convolutional neural networks into a unified model which is more robust to partial occlusions.
>
> [1] Wang, Xinlong, et al. "Repulsion loss: Detecting pedestrians in a crowd." In CVPR 2018.
>
> [2] Zhang, Shifeng, et al. "Occlusion-aware R-CNN: Detecting pedestrians in a crowd." In ECCV 2018.
>
> [3] Liu, Songtao, et al. "Adaptive nms: Refining pedestrian detection in a crowd." In CVPR 2019.
>
> [4] Kortylewski, Adam, et al. "Compositional convolutional neural networks: A deep architecture with innate robustness to partial occlusion." In CVPR 2020.
>
> [5] Chu, Xuangeng, et al. "Detection in crowded scenes: One proposal, multiple predictions." In CVPR 2020.
>
> [6] Wu, Jialian, et al. "Temporal-context enhanced detection of heavily occluded pedestrians." In CVPR 2020.
>
> [7] Ke, Lei, et al. "Deep Occlusion-Aware Instance Segmentation with Overlapping BiLayers." In CVPR 2021.

---

### Decision · Program_Chairs · 2021-10-09

**Decision:**

Accept

**Comment:**

The paper focuses on instance segmentations in videos of scenes with severe occlusions. This is a very challenging task, and improving models' capabilities in such scenarios will lead to significant advances in general scene understanding. The authors proposed a dataset and an associated challenge for this task. All the reviewers appreciated the usefulness of the dataset and the depth of the provided analysis. The authors updated the manuscript to take into account reviewers' recommendations.